# A content analysis-based approach to explore simulation verification and identify its current challenges

**Christopher J. Lynch**[1☯*], **Saikou Y. Diallo**[1‡], **Hamdi Kavak**[2‡], **Jose J. Padilla**[1‡]

**1** Virginia Modeling, Analysis and Simulation Center, Old Dominion University, Suffolk, VA, United States of America, **2** Department of Computational and Data Sciences, George Mason University, Fairfax, VA, United States of America

☯ These authors contributed equally to this work.
‡ These authors contributed equally to this work.
* cjlynch@odu.edu

**Data Availability Statement:** All relevant data are within the paper and its Supporting Information files.

**Funding:** The author(s) received no specific funding for this work.

## Abstract

Verification is a crucial process to facilitate the identification and removal of errors within simulations. This study explores semantic changes to the concept of simulation verification over the past six decades using a data-supported, automated content analysis approach. We collect and utilize a corpus of 4,047 peer-reviewed Modeling and Simulation (M&S) publications dealing with a wide range of studies of simulation verification from 1963 to 2015. We group the selected papers by decade of publication to provide insights and explore the corpus from four perspectives: (i) the positioning of prominent concepts across the corpus as a whole; (ii) a comparison of the prominence of verification, validation, and Verification and Validation (V&V) as separate concepts; (iii) the positioning of the concepts specifically associated with verification; and (iv) an evaluation of verification's defining characteristics within each decade. Our analysis reveals unique characterizations of verification in each decade. The insights gathered helped to identify and discuss three categories of verification challenges as avenues of future research, awareness, and understanding for researchers, students, and practitioners. These categories include conveying confidence and maintaining ease of use; techniques' coverage abilities for handling increasing simulation complexities; and new ways to provide error feedback to model users.

## 1. Introduction

Verification facilitates the identification and removal of errors within the discipline of Modeling and Simulation (M&S) to increase credibility in the construction of a simulation. This process helps model builders and users to identify whether unexpected behaviors appearing within a simulation are manifestations of incorrect construction and helps to increase users' confidence through the absence of errors. As the discipline has evolved over the past 60 years, challenges have emerged, changed, and disappeared due to technological and methodological advances combined with expansion into new application areas. New challenges continue to

**Competing interests:** The authors have declared that no competing interests exist.

emerge for increasing users' and stakeholders' confidence in a simulation, for increasing scalability of verification techniques across model components, and for developing new methods for providing feedback to model users. An examination of how the characterization of verification has evolved can provide insight into the perspectives that drive verification and provide direction for future research.

Verification is the process of determining that a simulation is built correctly [1–3]. This requires examining simulations' structure, code, and behaviors to identify implementation errors. More precise definitions range from the "debugging of the logic and code" of the simulation [4], to "determining that a model, simulation, or federation of models and simulations implementations and their associated data accurately represents the developer's conceptual description and specifications" [5], to "substantiating that the model is transformed from one form to another with sufficient accuracy" [6–10].

Similar to conducting validation (e.g. determining that a simulation adequately reflects the modeled system [11–13]), the results of verification reflect the absence of errors within the constraints of the applied tests. Verification's myriad foci areas have included analysis methodology, sample sizes, and replication [14] to big data, cloud computing, and decision support [15]. Numerous methodologies and frameworks explicitly include verification [4, 6, 16–18] and have evolved to include new efforts, including verifying against conceptual and reference models [19, 20], integrating verification into the life cycle of modeling and simulation studies [3, 21, 22], checking the simulation's experimental design [23]. The presence of an error may reveal itself through repeated occurrence across numerous runs. Occurrences that appear in a low percent of runs may not reflect an actual error. Repeated testing across the simulation' solution space is required to make this determination. Therefore, confidence increases by testing larger portions of the solution-space [24] and examining the histories of the simulations [25] leading to suspicious outcomes.

The challenges of differentiating errors from unexpected behaviors and in tracing events, interactions, and outcomes back to the underlying model specifications will continue to increase when dealing with systems of dynamic structure [26–28], human behaviors [29, 30], initialization using unstructured data [31, 32], and network homophily across multiple dimensions [33, 34]. Additionally, new challenges may emerge from continued methodological and technological advances, such as dynamically allocating computational loads at runtime [35], building and running simulations on the web [36–39], constructing hybrid simulations [40–42], and endeavoring to lower the barrier of entry for M&S to encourage STEM research [37, 39, 43].

Several surveys have explored the use of verification in practice. A model builder survey identifies verification as a largely trial and error activity with most respondents opting for informal verification [44]. Surveys of Swedish companies [45] and Australian companies [46] identify that software developers sparsely use structured approaches, instead using informal unit tests. Surveys of concurrent programs [47] and software developers [48] identify that object-oriented and unit testing are the most common techniques alongside code inspections and walkthroughs. Collectively, these surveys agree that verification is commonly de-prioritized and neglected due to time pressure and that people often opt for the easier to use informal techniques. These surveys suggest that the available approaches may not be getting used in practice because they are difficult to learn, their benefits are not clear, they are too time consuming to be applicable, and researchers are unaware of techniques outside of their own domains.

Several studies have examined the historical progression of verification due to computational advances [49], contributions of individual researchers [50], and tactile concerns [14, 15]. In practice, verification suffers from time and resource requirements [7, 45, 46, 51, 52],

familiarity and history of use [8, 45], and ease of use and learning curves [44, 46, 51]. Despite its significance and a perceived loss in simulation credibility when not conducted [53], verification is considered a lacking resource among modelers [7, 8, 54]. The multitude of terminologies, concepts, techniques, and methodologies along with their inconsistent definitions over time reduce clarity for verification [6, 50, 55–57]. This is further compounded by challenges resulting from: modelers having incorrect, incomplete, or contradictory knowledge about the system; disagreement among stakeholders on how to solve the problem or which problem to solve [58, 59]; conflicting baseline characteristics within the model [60]; simplifying the set of system behaviors; difficulties in examining the combinations and sequences of events leading to system level outcomes [61]; and model components changing over time [62, 63].

Understanding how verification has changed over time is important in facilitating healthy growth of the M&S discipline and in identifying avenues for conducting research. To this end, we conduct an exploratory study to examine how the concepts, defining characteristics, and foci of verification has evolved over the past six decades. Our aim is to illuminate semantic changes to verification as a term and to highlight that the primary role of verification is constantly evolving to meet and address new challenges that emerge in the development, exploration, and sharing of simulations. We then take our analysis a step further to compare the current characterization of verification with published literature within the past decade to highlight the current challenges for verification research. This study does not explore the semantic evolution of validation or V&V as individual terms. Nor does this study provide a comparative evaluation between verification, validation, and V&V.

We use content analysis to objectively evaluate the evolution of verification based on published, peer-reviewed literature comprising a corpus of 4,047 publications spanning the last six decades. Based on this analysis we identify and support the existence of three primary categories of verification challenges. Section 2 provides our methodology for constructing the corpus and conducting our analyses. Section 3 provides a discussion of our results. We discuss the current challenges and future research directions in Section 4 and conclude in Section 5.

## 2. Methodology

To investigate the role of simulation verification our methodology consists of the following steps: (1) constructing a Corpus that provides relevant Modeling and Simulation articles; (2) reducing this Corpus to a subset that specifically pertains to verification (resulting in a Verification Corpus); (3) grouping articles by the decades in which they were published; (4) conducting content analysis on the grouped Verification Corpus as a whole; (5) conducting content analysis separately on each decade; (6) gathering insight into the evolution of verification; and (7) identifying existing challenges to verification. Fig 1 displays our methodology.

*Step 1*: We establish our initial Corpus by collecting articles that specifically deal with simulation topics, methodologies, theories, tools, or applications across academia and industry. Initially, this Corpus contains 20,905 publications between the years of 1963–1965 and 1968–2015 in pdf, doc, and docx file formats. For the 4,424 PDF files obtained that were not machine-readable, we used optical character recognition (OCR) to convert them into machine-encoded text documents. Table 1 displays the publication sources, the range of publication dates for each source, the type of venue, and the total number of publications. For sources without free access to all of their publications, we obtained only the publications that were freely available that contained the term "verification". These publications are indicated by a "*" symbol within the fourth column of the table.

The selected journal venues cover a wide range of simulation topics that include both American and European perspectives. *SIMULATION*: *Transactions for the Society for Modeling*

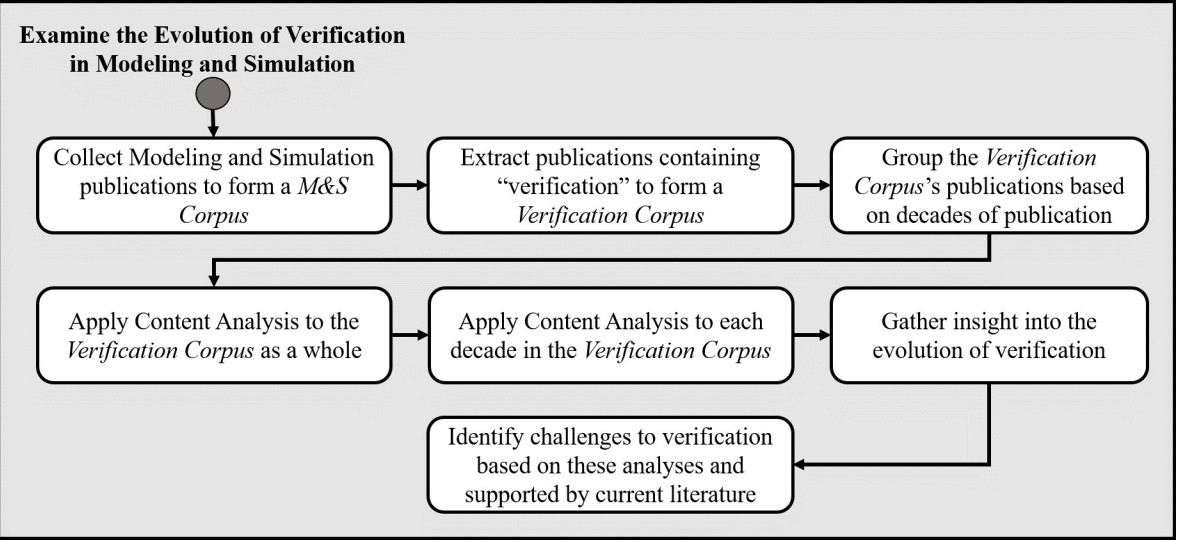

**Fig 1. Methodology for forming the Verification Corpus and conducting analyses.**

*and Simulation International* and *Simulation Modelling Practice and Theory* focus on advancements within Modeling and Simulation; *System Dynamics Review* focuses on advancements in system dynamics towards societal, managerial, and environmental problems; *Journal of Simulation* focuses on research within the fields of discrete-event simulation, system dynamics, and agent-based modeling; *Journal of Artificial Societies and Social Simulation* maintains an interdisciplinary focus on the exploration and understanding of social processes; and *Journal of Defense Modeling and Simulation* maintains focus on simulations for

**Table 1. Corpus constructed to conduct the content analysis study.**

| Venue | Venue Type | Year Range Retrieved | Total Articles Obtained | Subset of Articles containing the term "Verification" |
|---|---|---|---|---|
| **System Dynamics Review** | Journal | 1996–2014 | 23* | 23 |
| **Journal of Simulation** | Journal | 2007–2015 | 45* | 45 |
| **Journal of Artificial Societies and Social Simulation** | Journal | 1998–2015 | 666 | 92 |
| **Journal of Defense Modeling and Simulation: Applications, Methodology, Technology** | Journal | 2004–2015 | 276 | 79 |
| **SIMULATION: Transactions of the Society for Modeling and Simulation International** | Journal | 1963–1965 | 375 | 12 |
| **Simulation Modelling Practice and Theory** | Journal | 1994–2015 | 317* | 317 |
| **Winter Simulation Conference** | Conference | 1968–1971, 1973–2014 | 8933 | 1776 |
| **Simulation Interoperability Workshop** | Conference | 1997–2014 | 4012 | 936 |
| **I/ITSEC Conference** | Conference | 1968–2014 | 77* | 77 |
| **System Dynamics Society** | Conference | 1976–2014 | 5179 | 401 |
| **Computer Generated Forces–BRIMS Conference** | Conference | 1997–2008 | 557 | 88 |
| **System of Systems Engineering Conference** | Conference | 2006–2015 | 128* | 128 |
| **Spring Simulation Conference** | Conference | 2014–2015 | 317 | 73 |
| **Total Papers** | | | 20,905 | 4,047 |

* Indicates that only freely available papers were obtained.

military and defense. The selected conferences provide perspectives from modeling and simulation researchers, practitioners, students, and members of the industry with emphases on engineering, science, theories, methodologies, and applications. The venues are listed in column 1 of Table 1.

*Step 2*: We take the initial Corpus and refine it to include only publications which contain the term "verification" anywhere within their text. This reduced the Corpus to 4,047 publications including 653 publications that were OCR-converted PDFs. The final column of Table 1 provides the total number of publications from each venue comprising the refined Verification Corpus. Note, this simple filtering approach may yield articles that contain the term verification only within their reference lists or author biographies. A benefit of automated content analysis is the assignment of a category for the baseline of the analysis. Since concepts are examined with respect to their relative co-occurrence of the specified category, the remainder of the each text is ignored. Therefore, text within these articles that does not mention verification does not impact the frequency or prominence scores of the identified concepts. The Verification Corpus may provide a slight over-approximation of the number of verification articles, but these do not bias the analysis. Therefore, we do not apply additional manual effort for identifying and removing any such articles from the Verification Corpus.

*Step 3*: We organize the publications within the Verification Corpus based on their decades of publication. Table 2 displays the number of publications, the range of years, and the number of venues represented within each decade.

*Steps 4 and 5*: We use content analysis to automatically search the Verification Corpus and generate quantitative outputs. Content analysis creates content categories from a body of text to identify the main themes and ideas in a systematic and objective manner [64–67] for interpreting the text as a whole [68]. This technique produces a ranked list of concepts for each decade as well as a thesaurus that provides a list of terms frequently associated with each concept that is not frequently associated with other concepts within the corpus.

We select Leximancer to conduct our analysis as it is a robust tool to identify concepts from text and to calculate *relative frequency*, *strength*, and *prominence* values for every concept [69]. *Relative frequency* provides the conditional probability of a concept ($C_O$) given a category ($C_A$) (i.e., how likely is it that the concept will be mentioned given the particular category) as shown in Eq 1. Categories are identified as concepts that are highly interconnected to other concepts within the corpus. For our study, we select *verification* as the only category; thus, all *relative frequency*, *strength*, and *prominence* values are calculated with respect to only this category. *Strength* provides the conditional probability of a category ($C_A$), such as *verification*, given a particular concept ($C_O$), such as *data*, (i.e., the strength of the association between the category and the concept) as shown in Eq 2.

$$P(C_O|C_A) = \frac{P(C_O \cap C_A)}{P(C_A)} \tag{1}$$

$$P(C_A|C_O) = \frac{P(C_A \cap C_O)}{P(C_O)} \tag{2}$$

*Prominence* is a function of the strength and frequency for each concept/category combination and reflects the importance of a concept within a specific category. Formally, prominence is the joint probability of a concept A ($Con_A$) given a category C ($Cat_C$) over the product of marginal probabilities of the appearance of *A* and *C* within the text as shown in Eq 3. S1 Data

**Table 2. Verification Corpus organized by decade.**

| Decade | Years Included | Number of Publication Venues Represented | Number of "Verification" Publications | All Publications |
|---|---|---|---|---|
| 1960s | 1964, 1965, 1968, 1969 | 3 | 27 | 500 |
| 1970s | 1970–1979 | 3 | 100 | 799 |
| 1980s | 1980–1989 | 3 | 253 | 1,584 |
| 1990s | 1990–1999 | 8 | 822 | 5,018 |
| 2000s | 2000–2009 | 11 | 1,724 | 8,353 |
| 2010s | 2010–2015 | 11 | 1,121 | 4,651 |
| Total Publications | | | 4,047 | 20,905 |

displays the Leximancer configuration used for this study.

$$Prominence(Con_A, Cat_C) = \frac{\left(\frac{(co-occurrence\ count\ for\ A\&C)}{total\ number\ of\ context\ blocks\ in\ data\ set}\right)}{\left(\frac{occurrence\ of\ A}{number\ of\ context\ blocks}\right) * \left(\frac{occurrence\ of\ C}{number\ of\ context\ blocks}\right)} \qquad (3)$$

In Step 4, we conduct automated content analysis on the Verification Corpus. We use the folder groupings from Step 3 to tags each concept with their publication decade which results in the same set of concepts being scored in each decade and for the corpus as a whole. As a result, it becomes possible to track the existence and prominence of concepts within different decades. We then extract the list of concepts for each decade along with a thesaurus of terms that span the entire Corpus.

In Step 5, we conduct six separate content analyses using only the groups of publications within each decade. This provides a unique set of prominent concepts for each decade. We extract the concepts associated with verification along with each decade's thesaurus. We utilize these thesauri to evaluate each decade's defining characteristics with respect to verification.

*Step 6*: Finally, we gather insight into the evolution of verification from five directions. First, we utilize the thesauri to examine the terms that are frequently found in context with verification that are not found as frequently alongside other concepts within the Verification Corpus. Second, we establish a baseline view of each decade by examining this Corpus as a whole with publications tagged by their decades of publication. Third, we identify the prominent concepts with respect to verification unique to each decade. Fourth, we identify how the definition of verification changes over time. Finally, we explore the effect of the appearance of the term V&V on the prominence of verification. Table 3 summarizes our explorations to gather insight with respect to the method and corpus used.

**Table 3. Summary of the main goals, analysis method, and the component of the corpus used to provide the necessary data for Step 6 of the methodology.**

| Goal–Gain Insight into: | Method | Corpus Component |
|---|---|---|
| Identify the evolution of the Verification Corpus's concepts (Section 3.1) | Extract the Verification Corpus's ranked concept list with each publication within the Corpus tagged by its decade of publication. Compare concept rankings across decades. | Verification Corpus |
| Identify prominences of verification, validation, and V&V over the decades (Section 3.2) | Compare the prominences of *verification*, *validation*, and *V&V* using the ranked concept lists generated from the Verification Corpus's tagged publications. | Verification Corpus |
| Identify the evolution of verification's concepts over the decades (Section 3.3) | Extract and analyze the ranked concept list for each decade's content analysis conducted using only the publications within that decade. | Verification Corpus divided into six decades using "verification" as the comparison category |
| Identify the defining characteristics of verification each decade (Section 3.4) | Using the thesaurus generated from each decade's content analysis, analyze the terms commonly associated with verification. | Verification Corpus Thesauri from each of the six decades using "verification" as the comparison category |
| Identify challenges and future directions for simulation verification research (Section 4) | Explore common themes within the concepts and definitions pertaining to verification over time to identify existing challenges. Then, conduct a literature review to reflect the state-of-the-art. | The evolution of concepts and defining characteristics identified from Sections 3.3 and 3.4 |

*Step 7*: We then identify challenge areas based on the insights gathered of the defining characteristics from Step 6 and the concepts identified from Step 5. To reflect the current state-of-the-art of these challenge areas, we reinforce our discussion with further reviews of existing verification literature published within the past decade. These challenge areas point towards open areas for innovation, development, and creation of future research for verifying simulations and advancing simulation verification processes.

## 3. Results

We generate results in a top down progression starting with an overview examination of the Verification Corpus's prominent concepts (not categorized under verification) to gain insight into the foci of the verification literature each decade. Then, we compare the prominence values of verification, validation, and V&V each decade to gain a baseline understanding of how prominent verification is with respect to both of these concepts. Next, we examine the Verification Corpus's prominent concepts when classified with respect to verification to gain specific insights into the evolution of verification over the decades. Finally, we examine the thesaurus of verification's related terms in each decade to gain an understanding of the evolution of verification's defining characteristics.

Roughly 19.36% (4,047 of the 20,905 articles) of the overall articles pulled from the selected M&S venues form the Verification Corpus. However, when looking at the venues which contributed papers that were did not just contain verification this percentage drops to 17.02% (3,457 out of 20,315 articles). This is not necessarily a negative observation as these articles range from M&S applications and case studies to theory, methodology, and technology. A one in five reporting rate may be due to a lack of standardized approaches for conducting and reporting verification activities across modeling paradigms and methods. This finding supports the argument that verification is underutilized within the community and that simpler means for conducting and communicating its results are needed (7, 8, 44, 54).

### 3.1 Evolution of concepts across the Verification Corpus

We conduct an analysis of the Verification Corpus using all decades simultaneously with each publication tagged with its decade of publication. This ensures that all concepts within the Verification Corpus are examined with respect to their prominence within the corpus as a whole. Frequency, strength, and prominence values are calculated with respect to all of the text in order to normalize the prominence scores within the Verification Corpus. Concepts that do not appear within certain decades (i.e. *V&V*) result in prominence values of zero during those decades. A total of 102 prominent concepts appear. S2 Data contains the complete list of ranked concepts for each decade along with their prominence scores.

The formation of the concept list includes each concept that has a prominence score of at least 1.0 within any decade. This results in concepts showing prominence scores less than 1.0 within various decades, but always greater than 1.0 in at least one decade. For exploration, we examine the concepts comprising the top 10% of each decade within the corpus as these provide insights into the main topics each decade. Table 4 displays the top 10% of prominent concepts. The prominence rankings are used to explore the concepts in each decade. Prominence scores are relative to "verification" as the specified category and are based on the total occurrences of each related term each decade. As a result, the range in the magnitude of concept prominences is expected to differ each decade.

In the 1960s and the 1970s, the two most prominent concepts are *computer* and *program*. These concepts reinforce a focus on the executable version of the model. These concepts pertain to what is going into and out of the simulation, the values (types and ranges) that a

**Table 4. Verification Corpus's top 10% most prominent concepts.**

| 1960s | | 1970s | | 1980s | |
|---|---|---|---|---|---|
| **Concept** | **Prominence** | **Concept** | **Prominence** | **Concept** | **Prominence** |
| computer | 5.638 | computer | 2.640 | computer | 1.859 |
| program | 2.830 | program | 2.569 | program | 1.626 |
| group | 2.500 | total | 1.829 | input | 1.479 |
| function | 2.474 | distribution | 1.572 | output | 1.419 |
| output | 2.111 | simulated | 1.527 | production | 1.281 |
| social | 2.058 | rate | 1.448 | techniques | 1.214 |
| total | 1.838 | function | 1.439 | system | 1.193 |
| distribution | 1.773 | input | 1.386 | form | 1.185 |
| input | 1.718 | population | 1.358 | problem | 1.171 |
| value | 1.625 | output | 1.331 | structure | 1.166 |
| **1990s** | | **2000s** | | **2010s** | |
| **Concept** | **Prominence** | **Concept** | **Prominence** | **Concept** | **Prominence** |
| object | 1.810 | HLA[a] | 1.527 | social | 1.711 |
| HLA | 1.373 | V&V | 1.302 | population | 1.680 |
| interface | 1.329 | M&S | 1.262 | agent | 1.593 |
| program | 1.287 | architecture | 1.249 | policy | 1.452 |
| event | 1.231 | capabilities | 1.220 | power | 1.393 |
| execution | 1.222 | technology | 1.207 | average | 1.387 |
| verification | 1.182 | training | 1.197 | total | 1.354 |
| test | 1.182 | distributed | 1.193 | dynamics | 1.338 |
| computer | 1.173 | component | 1.179 | algorithm | 1.304 |
| software | 1.172 | effort | 1.176 | study | 1.298 |

[a]The High Level Architecture (HLA) is an IEEE Modeling and Simulation Interoperability Standard developed by the Defense Modeling and Simulation Office (DMSO) and adopted by NATO [70]. The HLA facilitates specifying and exchanging information when creating a simulation by federating simulations.

simulation uses and produces, and ensuring that these components are correctly contained within the executable simulation. Additionally, the concepts of the 1970s display a shift in focus towards the ability to conduct distributed simulation exercises across computer networks.

In the 1980s, focus shifts towards the static components of a simulation, including the *system* being modeled, the *problem* being addressed, and the *structure* of the simulation. Concepts focusing on the production of data (i.e. distributions and functions) no longer appear in the top 10%. While these concepts largely focus on model structure, they still fall within the overarching focus on the executable computer program.

Starting in the 1990s we witness a rapid change in prominent concepts. *Computer* and *program* have been dislodged from the top-two positions. *Verification* is one of the most prominent concepts of the 1990s and this is the only decade where both *verification* and *test* appear in the top 10%. This decade focuses largely on implementation-specific concepts. *Interface* emphases focus on implementing human-computer interactions for constructing simulators or allowing user participation inside of simulations.

In the 2000s, *computer* and *program* disappear from the top concepts while the HLA becomes the most prominent concept. *Verification* also disappears from the top 10% and is replaced with the concept of *V&V*; thereby representing the shift from separately verifying and validating simulations into a joint construct within the modeling process. The main focus highlights the advancement of the theory and the science of Modeling and Simulation as its

own discipline. *Training* reflects a critical usage of simulations which can suffer from serious repercussions if not properly supported by verification.

In the 2010s, *computer* and *program* now reside within the bottom 10% of the 102 concepts. The focus shifts to the applications areas of Modeling and Simulation, including *social* systems, *population* modeling, *policy* modeling, and *power* modeling. *Dynamics* of systems and outputs are prominent features with specific attention given to simulation outputs with respect to *averages* and total *values*.

Overall, initially prominent concepts that fit with the traditionally accepted definition of verification (i.e. *computer*, *program*, *input*, and *output*) lose prominence over time. To examine how closely related each decade's prominent concepts are to each other, we calculate the Pearson correlation coefficients for the prominence values across decades. Correlations close to +/-1.000 indicate strong positive or negative relationships between concept prominences across decades while correlations closer to 0.000 indicate weak relationships. Table 5 displays the correlations of prominence values between each decade.

These correlations reflect that the perception of all 102 of the verification concepts are relatively consistent from the 1960s through the 1980s. The 1960s, 1970s, and 1980s are highly correlated with each other, indicating a relative stability in the perceived importance of concepts throughout this period. A change occurs in the 1990s that alters the perception of important concepts, as indicated by the lack of correlation with the prior decades. This change corresponds with the appearance of V&V as a concept in the 1990s and may indicate a loss of specific focus on verification due to the merging with validation into "V&V". Furthermore, the lack of correlation of the 2000s and 2010s with any other decade may indicative the lack of a cohesive perception across the community on how to conduct or communicate verification within their publications. Alternatively, this may reflect that new challenges, topics, or techniques for conducting verification have emerged with have resulted in the creation of new concepts or the resolution of previous concepts. A more in-depth look into each decade is needed to examine how verification evolves throughout this period. We conduct this analysis in sections 3.4 and 3.5.

### 3.2 The prominence of V&V, verification, and validation over the decades

Following the dominating prominence of validation alongside verification beginning in the 1980s along with the strong intercoupling of the two terms within current verification literature, we explore the changing prominences of the concepts *verification*, *validation*, and *V&V*. As a merged concept, *V&V* does not appear until the 1990s with a prominence of 0.0 showing for the 1960s, 1970s, and 1980s. During the 1990s *V&V* is less prominent (1.037 prominence) than *verification* (1.182 prominence) but more prominent than *validation* (0.982 prominence). However, during the 2000s *V&V* becomes more prominent than both *verification* (1.074) and *validation* (0.984) with a prominence of 1.302. In the 2010s, *validation* beats out the other

**Table 5. Correlations of each decade's concepts prominence values. Color intensity indicates the strength of correlation (green is positive and red is negative).**

| Decade | 1960s | 1970s | 1980s | 1990s | 2000s | 2010s |
|---|---|---|---|---|---|---|
| 1960s | - | 0.749 | 0.558 | 0.028 | -0.458 | -0.033 |
| 1970s | | - | 0.704 | 0.007 | -0.663 | 0.064 |
| 1980s | | | - | 0.225 | -0.633 | -0.124 |
| 1990s | | | | - | 0.196 | -0.855 |
| 2000s | | | | | - | -0.556 |
| 2010s | | | | | | - |

terms with *V&V* dipping and *verification* reaching its all-time low. Fig 2 displays the *verification*, *validation*, and *V&V* prominence values during each decade. This reflects a correlation between the appearance of V&V in the 1990s and a decline in prominence of verification over the following decades.

The importance of the *verification's* prominent position over that of validation in the 1960s - 1980s is an indication that verification received more attention within the texts that mentioned both verification and validation. The closeness of the three prominence values from the 1990s through the 2010s may be a good indicator that publications are beginning to devote equal amounts of space to both verification and validation or it may indicate that verification and validation are equally receiving less attention. We take this as a promising sign that when both terms are mentioned within a text that *V&V* is intended to convey both the correctness of the implementation along with how well the simulation reflects the modeled system.

### 3.3 Evolution of verification concepts over the decades

We examine the temporality of the prominent concepts pertaining to verification using only the publications within each decade. S3 Data provides the ranked concept lists and their corresponding prominence values pertaining to "verification". Since verification always has the highest prominence with respect to itself within each decade, we exclude it from the following lists and analyses.

We construct word clouds to visualize the varied foci of verification over time. Fig 3 provides a timeline-ordered configuration of these word clouds with each decades' concepts spelling out the word "verify". The concepts within each letter reflect the corresponding decade only. The size of each term reflects the relative frequency of that concept with respect to verification within the corresponding decade. The specific values underlying the terms portrayed in Fig 3 are provided within S3 Data under the column headers of "frequency with verification" next to each decade's concept list.

The word clouds for the 1960s and the 1970s internally contain fairly evenly sized words which reflects evenly weighted prominence values across both decades. However, with the appearance of validation in the 1980s all of the other concepts begin to be overshadowed by the volume of co-occurrences of the term validation. This effect can be clearly observed through visual inspection. In the 2000s, validation has become so large that the other terms are very hard to discern. While the other terms are becoming easier to discern within the 2010s,

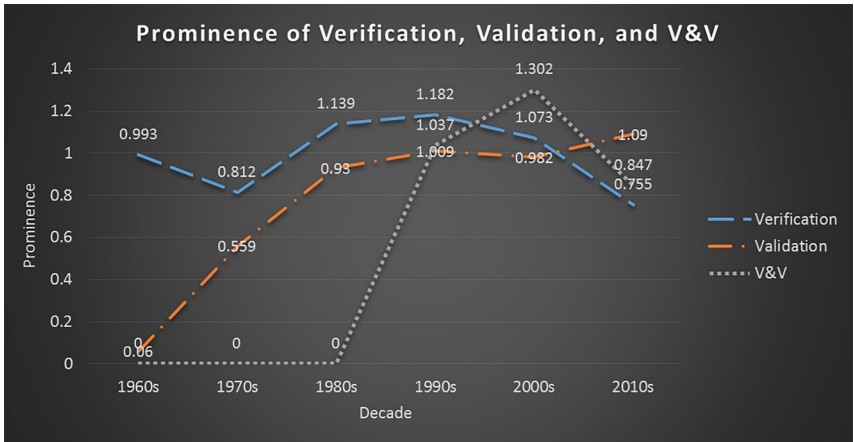

**Fig 2. Prominence values of verification, validation, and V&V from the 1960s to the 2010s.**

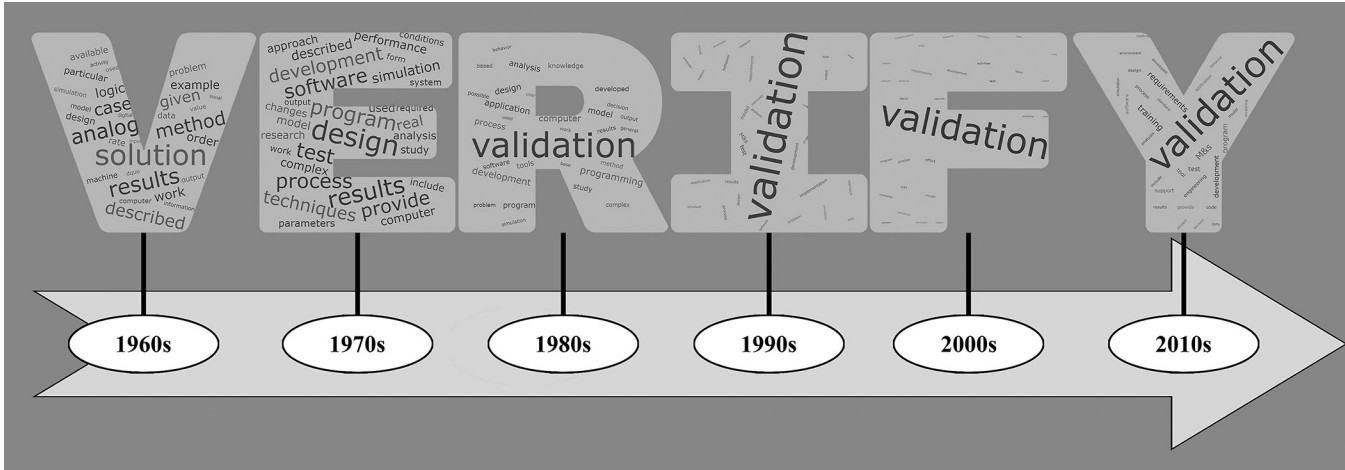

**Fig 3. Word clouds of concepts that frequently occur with verification each decade with word size indicating relative frequency within the corresponding decade.**

some of the more legible concepts are starting to reflect the definition of validation instead of verification, such as training and support. This indicates an increasing importance on verification techniques to directly support validation efforts. Table 6 provides the top 10% of concepts each decade ranked by prominence.

**Table 6. Top 10% of prominent verification concepts obtained through independent content analyses.**

| 1960s | | 1970s | | 1980s | |
|---|---|---|---|---|---|
| Concept | Prominence | Concept | Prominence | Concept | Prominence |
| solution | 8.449 | design | 6.072 | Validation | 10.795 |
| results | 6.331 | results | 5.188 | Programming | 2.980 |
| analog | 6.075 | program | 4.559 | Development | 2.912 |
| method | 5.503 | test | 4.518 | Computer | 2.834 |
| described | 5.043 | process | 4.497 | Application | 2.700 |
| case | 4.950 | software | 4.364 | Model | 2.634 |
| given | 4.455 | provide | 4.247 | Process | 2.623 |
| logic | 3.819 | development | 3.995 | Design | 2.551 |
| work | 3.819 | techniques | 3.869 | Analysis | 2.549 |
| order | 3.457 | real | 3.240 | program | 2.424 |
| **1990s** | | **2000s** | | **2010s** | |
| Concept | Prominence | Concept | Prominence | Concept | Prominence |
| validation | 20.908 | validation | 25.043 | validation | 17.138 |
| M&S | 2.933 | effort | 2.577 | M&S | 4.163 |
| test | 2.858 | model | 2.253 | training | 4.026 |
| model | 2.535 | development | 2.243 | requirements | 3.831 |
| development | 2.153 | test | 2.237 | program | 3.280 |
| implementation | 2.112 | M&S | 2.229 | development | 3.116 |
| results | 2.068 | results | 2.221 | test | 3.010 |
| requirements | 2.049 | requirements | 2.199 | tool | 2.486 |
| behavior | 2.018 | process | 2.186 | engineering | 2.384 |
| process | 1.983 | activities | 2.122 | support | 2.381 |

The 1960s maintain focus on the executable model with sub-themes, such as *logic* and *method* combined with *solution* and *results,* pointing towards stronger relationships for producing results. The 1970s reveal a stronger focus on the importance of model design. Concepts focus on design, exploring the program and the software, along with connections to tests, processes, and techniques used in verifying these components. The 1980s serves as the starting point of *validation* as the single most prominent concept associated with *verification.* In the previous two decades, prominence values are relatively close to each other reflecting a shared level of importance between the concepts. The term validation now occurs so often alongside verification that validation has become the dominating concept within the rankings.

The 1990s reveal an almost doubled prominence in validation over the previous decade. *Behavior,* commonly associated validation, now appears as a prominent concept and reflects a change in focus to testing simulation behaviors against the expected behaviors of the system. Verification is often mentioned in the company of *M&S* which emphasizes its perceived importance within the discipline. The prominence of *validation* reaches its peak in the 2000s. Verification is now 10 times more likely to connect with validation than with the next most prominent concept. The magnitude of the difference between the verification-validation prominence value and the other top concepts reflects the strength of the bond formed within the corpus and the blurring of verification's identify within the publications. *Effort* appears as the second most prominent concept and reflects the dependencies on time, money, and other resources.

The 2010s highlight the importance of continued verification education and training for modeling and simulation professionals. For models and simulations intended for use as *support*, *tools*, *engineering*, or *training*, it is critical that they undergo verification to ensure that they are developed correctly and meet their baseline requirements before being used. Validation remains the most prominent concept. Additionally, the presence of concepts that are traditionally associated with validation, such as training and support, represent a shifting perspective towards validation.

## 3.4 The defining characteristics of verification each decade

The prominent concepts identified within 3.3 provide unique looks into the perception of the community on verification over time. Additionally, we want to get an idea for the definition associated with verification in each decade in order to observe changes in the definition. Definitions are made up of three parts, including the term, the class pertaining to the term, and its distinguishing characteristics [71]. For this examination, the term is "verification" and its class is "a process". We agree that verification is the process of making sure that the simulation is built correctly; however, we seek to identify its distinguishing characteristics within each decade.

We utilize the thesauri generated from the independent content analyses to illuminate the characteristics that comprise the meaning of verification. We exclude proper nouns, partial and misspelled words, and mathematical terms from our analysis. The terms are listed starting with the highest ranked within the thesaurus and proceed in descending order. S4 Data provides the complete thesaurus of terms pertaining to the concept "verification" from each decade and their corresponding values.

- *1960s* Terms include accounting, all-digital, advocates, assertion, coding, computer-asked, discover, elaboration, harder, impossibility, influential, investigating, objection, precautions, pretend, shooting, and spending.

These terms define *verification* as a discovery process that is application-oriented and specifically deals with digital computer code. A primary aspect involves checking assertions and

alleviating cautions against the use of a simulation. Validation does not appear as a ranked term under verification.

No specific instruction is provided on when to conduct verification tasks. With the emphasis on coding and the computer, code-based feedback appears to be the most common indicator of errors in this decade.

- *1970s* Terms include assurance, mathematical-logical, aleatory, and-validation, civilian, compile-time, concurrency, consultants, digital, fault-free, in-program, machine-oriented, multidisciplinary, non-synchronous, post-run, programmer-oriented, routinely, stuck-at-(zero or one), designer's, and multi-stage.

These terms define *verification* as a process of assuring users that a simulation works as intended and is very reflective of software engineering practices. There is a clear association with verifying a simulation's mathematical, logical, programmatic, and hardware components. Analyses appear to be primarily concerned with conducting verification either before a simulation run starts or after it completes. Code appears to be the primary target of verification activities. Validation now appears in connection to verification. No simulation languages or formalisms appear and nothing specifically connects to hybrid or multi-paradigm models. This reinforces the claim that verifying hybrid models remains an open question [40, 42].

A focus on conducting verification after the simulation is completed running (post-run) as well as at compilation (pre-run) is identified to explore errors pertaining to both the mathematical and logical components of the simulation. Modelers focus on identifying errors due to concurrency and look for areas where the model gets stuck on constant values. Verification feedback appears to heavily rely on numerical indictors to convey errors.

- 1980s Terms include substantiating, validates, breakpointing, non-simulators, positivism, accountability, behavior-mode, codify, computer-executable, deception, developer's, empiricists, falsificationist, histograms, instrumentation-based, non-simulationists, simulation-related, walk-throughs, and substantiation.

These terms define *verification* as a process that ensures non-simulation experts can rely on a simulation's outputs as well as a collection of techniques for verifying correctness, such as creating histograms or conducting walkthroughs. Verification is connected to validation and depends on the simulation's behavior compared against empirical evidence. The appearance of positivism and empiricism reflect that driving schools of thought include verifying each of the model's assertions.

The verification process relies heavily on computer science techniques for checking code and model structure. Techniques gain a focus on applying to the simulation during execution and adds a requirement for instrumenting the simulation to provide the necessary data over time. This indicates a combination of high simulation, statistical, and computer science knowledge requirements on the person conducting verification. Feedback remains largely code-based, but now also includes histograms for visualizing distributions using continuous data.

- *1990s* Terms include validation, assuring, empiricism, substantiation, rationalism, structure-oriented, substantiating, multi-step, methodical, post-construction, reproducible, software-in-the-loop, approves, ascertaining, inconvenient, confrontational, critics, post-condition, simulation-model, specificity, undocumented, audits, beta-testing, clarifies, data-input, deductions, methodically, non-modelers, positivism, specification-oriented, unproductive, visually-based, behaviors, bug-free, checkers, cross-validation, divergence, easy-to-understand, laborious, peculiarities, situation-specific, t-statistic, unachievable, variability-sensitivity, and multistage.

   

These terms define *verification* as a methodical process applied after the simulation construction that focuses on providing evidence to critics that the simulation is free of bugs. A number of specific tests are associated, including code audits, beta-testing, and hypothesis testing using the t-statistic. While many of these tests are code specific, their metrics are situation specific and deal with behaviors and divergence of simulated outputs against expected outcomes. Verification is confronted with negative connotations as a result of completely verifiable simulations being unachievable, inconvenient, and confrontational. This problem corresponds to high needs for data storage and analytical platforms that can process constantly increasing volumes of simulation data [15].

Visually-based and easy-to-understand both appear as prominent characteristics. This reflects a desire for intuitive feedback with easily sharable results; however, this also fits historically with the increasing ease in developing visual representations of data that computers are yielding during the 1990s. The application of verification expands into new schools of science, including rationalism and positivism.

The verification process targets model structure after the construction of the simulation. Techniques now focus on comparing the simulation against its specifications as a metric for success or failure. There is also an indication that tests are designed to be situation specific which reflects the current perspective that V&V should test the model with respect to its intended purpose [72, 73]. Feedback mediums reflected include numerical for statistical measures, code-oriented for computer science tests, as well as visual.

- *2000s* Terms include validation, contingent, walkthroughs, ascertaining, formalization, conceptual-model, empiricism, generalizable, non-statistical, rationalism, relativist, truthfulness, accreditations, air-traffic, breakpoint, coder, comprehendible, disaggregating, domain-experts, evaluative, extreme-conditions, foundationalist, machine-readable, makefiles, multi-processor, negating, nonexistent, software-in-the-loop, specification-calibration, standardizes, sub-modules, terabyte, time-flow, tool-set, unsustainable, variability-sensitivity, advises, alleged, back-propagation, battlespaces, behavior-sensitivity, computerization, cost-risk, cost effectiveness, cross-element, cross-model, empirical-(strong/weak), interoperated, kilobytes, non-repeatable, structural-strong, structure-oriented, and voids.

These terms define *verification* as comparing a simulation to its conceptual model, its formalization, and its structural components in addition to its code. Focus shifts off of empiricism-based verification onto relativism and foundationalism techniques using non-statistical techniques. Concerns arise with respect to utilizing multiple processors and generating data in excess of terabytes. Decisions to conduct verification depend upon cost effectiveness considerations. Techniques are split between verification, such as code walkthroughs and utilizing break points, and validation, such as calibration and behavior sensitivity tests. This decade holds the largest emphasis on the costs associated with verification. A high presence of informal verification activities are revealed in this decade, as walkthroughs, face validation via subject matter experts, and extreme-conditions tests are generally selected *ad hoc* and conducted informally.

The instruction indicated within this decade reflects a strong emphasis on computer science techniques; however, model oriented techniques now also appear in the form of extreme conditions testing and behavior sensitivity testing. The emergence of techniques which focus on the model instead of its code reflects a maturation of M&S verification starting in this decade. Feedback mediums reflected include numerical for examining model sensitivity, visual for check model structure, and code-oriented.

- *2010s* Terms include accreditation, statechart-assertions, time-constrained, afterthought, execution-based, explainable, hides, inter-agent, non-bottlenecks, parameterization,

stressful, tenuous, abduction, evidence-driven, examine, flow-time, implausible, inaccessible, inexpensively, methodical, negated, non-simulation, painstaking, quickness, case-studies, computerized, decision-oriented, design-time, drill-down, intra-agent, satisfiability, trace-driven, and source-code.

These terms define *verification* as a critical resource for supporting and conveying credibility. Verification relies on examining the simulation while it executes and identifying areas of bottlenecks. Credibility relies on examining how model assertions hold throughout a run and exploring interactions between agents. Timing considerations for designing, conducting, and evaluating tests verification activities are very prominent within this decade. An indication of design of experiments is revealed through the presence of *parameterization* and *methodical*.

The instruction indicated within this decade reflects conducting verification in a methodical manner through model parameterization, conducting traces, and by producing evidence. There is a need reflected for feedback to support providing insight on execution; however, specific feedback for conveying what is happening during execution is not clearly discernable. Additionally, feedback mediums continue to reflect a focus on model structure and code.

## 4. Discussion: Challenges and future directions for simulation verification research

Throughout the decades the core essence of verification remained the same; that is, the goal is to convey a fault-free or error-free implementation of the computerized simulation. However, our evaluation illuminates changes to the primary testing targets, primary technique categorizations, and feedback mediums for communicating results over time. Targets of verification have relied on source code, model specifications, and defined structure. Technique categorizations have evolved from primarily code-oriented that are strongly rooted in software engineering or computer science (e.g. audits and walkthroughs) to model-oriented and simulation-oriented that reflect the maturation of the M&S discipline (e.g. extreme conditions testing) with considerations given to the correct correspondence identified between simulation structure and outcomes compared to the model design and known real-world values [20, 74].

The content analysis reveals a strengthening association to *when*, *where*, and *how* to conduct verification over time. *When* reflects the myriad of points that simulations undergo verification throughout the M&S process, such as by evaluating the simulation pre-runtime (e.g. static code), evaluating occurrences during execution, or evaluating outputs after simulation execution has completed. *Where* points to the simulation objects or data outcomes that facilitate evaluation, such as evaluating specific sub-modules within the simulation, exploring the source code, or testing specific model assertions. *How* relays the techniques and methodologies utilized to identify errors, such as evaluating state-chart assertions, using extreme conditions tests, or methods for parameterizing the simulation to facilitate the execution of a design of experiments.

No evidence is found of verification aspects existing that are specific to any given modeling paradigm, reinforcing conclusions presented by Sargent and Balci [50] and Brailsford, Eldabi [75]. This finding reflects the common practice of *ad hoc* selection and application of V&V techniques. A perspective shift could align the development of new techniques onto the underlying characteristics of the simulation to direct verification efforts. For instance, a common approach to verifying ABMs is to use traces to follow the execution of agents throughout execution to determine the correctness of the model's structure and accuracy of the outcomes of the interactions [61, 72]. This generic definition relies on a model's intended use to direct the trace examination. Alternatively, if the trace methodology was designed to instead focus on verifying time-based (i.e. an agent's dynamic age increase triggering a static change in life status) or resolution-based (agent, group, or population level) components within the simulation,

this could result in a more generalized and reusable approach for conducting trace verification. Ultimately, this could support reuse of techniques within modeling paradigms and contribute to more consistent reporting methods.

The changes to the defining characteristics of verification in each decade reflect technological and methodological advancements within the discipline [50] and occur as a result of technology advances creating new or solving existing problems [14, 15, 49, 76]. Verification costs increase as a result of increasing system autonomy, complexity, and abilities to assess their own status [77]. These characteristics reflect challenges pertaining to parallel execution, large amounts of simulated data [15, 78], building and running models in the cloud [38, 39, 60, 79, 80], and tracing the occurrences of errors to their sources [61, 81, 82].

From the evolution of concepts identified in Section 3.3 and the defining characteristics identified in Section 3.4, we identify the following three categories of verification challenges: (1) the ever-present need to increase user confidence while facilitating ease of use; (2) the need for increased coverage of verification techniques to handle increasing simulation complexity; and (3) the need to further investigate and contribute advances in feedback mediums for conveying verification results. Explanations and sub-categorizations of each of these challenges are explored in greater depth in the following three sub-sections and are supported by related, current verification efforts. We explore the current state-of-the-art with respect to each of these challenge areas by conducting a literature review of recent publications to identify their current challenges. The goal is to inform current researchers, students, and practitioners on existing challenges, to enable new researchers entering the domain of M&S, and to illuminate avenues for future verification research.

## 4.1 Conveying confidence and facilitating ease of use

The prominent concepts from the Verification Corpus indicate a sustained theme of conveying confidence and facilitating ease of use. As this is a driving premise of verification, this is expected within the analysis. However, the changing characterizations of verification over time reflect thematic differences within this theme. Concepts of *test*, *requirements*, *analysis*, *design*, *development*, and *results* reflect the traditional verification aspects of determining that a simulation has been implemented correctly. The defining characteristics across the decades reveal insight into shifting focal points within this theme, such as investigating (1960s), concurrency and fault-free (1970s), substantiating and accountability (1980s), substantiating and assuring (1990s), truthfulness and accreditation (2000s), and accreditation and explainable (2010s). The pursuit of confidence has ranged from searching for concurrency errors and identifying software faults to substantiating adherence to requirements and design to establish credibility and truth. Existing challenges pertaining to confidence and ease of use are explored further to assess the current state-of-the-art.

Model stakeholders' goals range from reducing costs within real world systems [83], to increasing reliability for training [84] and healthcare scenarios [85], to gaining insight into high risk or safety critical systems [86], as well as representing complex human behaviors [29, 87]. Verification supports these efforts by identifying or showing the absence of errors to increase a simulation's credibility and reliability [4, 6, 83] as well as its reuse [56]. Additionally, this helps to prevent Type I and Type II validation errors [4, 88]. Nance and Sargent [49] point out that users' abilities to recognize errors and to understand how outputs are produced are impaired when they are not part of the development process. Intuitive, transparent, and interpretable means for conducting verification and communicating results to non-experts are needed. Communicating reliability, credibility, and trustworthiness to lay users requires that verification results contain the following properties:

1. The results are easily understandable for the intended audiences

2. The results indicate all baseline testing assumptions, such as minimum sample size requirements, and convey that they are upheld without shifting the analytical burden onto the audience

3. The results reflect the testing conditions (e.g. the ranges of parameter values explored during testing as part of the design of experiments)

Current challenges for conveying confidence and maintaining ease of use in verification:

- **Communicating results to non-experts.** While several cloud-based simulation approaches support STEM education [37, 89] by making simulations more accessible to non-M&S experts, emphasis is needed for facilitating students' understanding of verification's roles in searching for errors as well as for establishing trust in the simulation. This process can be facilitated through the creation of online verification tools that can verify specific components or types of challenges. These tools and/or processes need to internally assess the adherence of the data being verified to the underlying assumptions of the tests being applied. Adherence or violations to these assumptions need to be clearly communicated alongside the results to further support credibility in the simulation outcomes. By connecting assumptions directly with pass/fail results of testing, increased power can be achieved from testing without requiring increased background knowledge of analytical techniques. Baseline descriptive and sampling statistics should help to connect with non-experts, as these techniques rely on rudimentary mathematics, probability, and statistics concepts [90].

Simulation environments, languages, and tools are becoming more prolific, easier to use, and more robust thanks to the World Wide Web. Modelers experience significant advances in simulation computing power through cloud computing [17]. Web-based model building represents a shift from model building to model assembling which requires new approaches for dynamically checking the implementation [36]. This allows modelers to build more complex simulations, build them faster, access them easily, make changes on-the-fly, and share them with wide audiences. Current cloud-based simulation challenges pertain to the interoperability and service architectures employed [38, 79, 91, 92]. As a result, greater V&V responsibilities thereby transfer to the providers of the simulation environments to provide the necessary capabilities for users to conduct tests that can reveal the presence of errors [38, 93].

- **Creating and extending tools to aid in verifying specific components of simulations.** Verification seeks to identify errors within an executable simulation; therefore, the processes, techniques, and metrics of failure change depending on the modeling paradigm, simulation platform, or programming environment used to construct the simulation. For instance, Lynch [94] identifies roughly 110 verifiable properties from just 9 different DES node types within the CLOUDES [39] simulation platform. A survey of academic, government, and industry M&S professionals found that informal verification techniques are most commonly utilized in practice with trial-and-error and visual inspection reported as the most common approaches [44]. Future research efforts can focus on investigating, designing, and creating new ways for simulation users to interact with simulation results in manners that potentially maintains the functional aspect of conducting trial-and-error exploration that relies on visual inspection. These tools should provide increased power in (1) identifying potential errors, (2) assessing adherence to test assumptions, and (3) mitigating mathematical, simulation, and statistical knowledge requirements.

- **Communicating adherence to a test's underlying assumptions, boundaries or requirements.** Analytical tests, whether being used for verification, validation, or exploration, have

underlying assumptions that must be met in order for the results to be valid. Therefore, when generating and communicating results of testing, the assumptions of the test need to appear with the results along with an indication of any violations in the test conditions. For example, statistical tests applied to data samples need to contain a minimum number of observations in order to presume that the sample is representative of its population. It is commonplace for many statistical packages and tools to include the sample size of the data included within a test function as part of the output along with a warning or error message if the sample size does not meet the needed requirements. This greatly assists the user in determining whether the outcome can be considered credible. This same practice should hold for all V&V and exploration tests applied to simulations.

## 4.2 Increasing coverage of techniques to handle increasing model complexities

The prominent concepts from the Verification Corpus indicate a theme of verification coverage with respect to model complexities. Concepts of *logic*, *software*, *techniques*, *process*, and *behavior* reflect differing levels and requirements of simulation space coverage in examining simulation models. Defining characteristics within the decades include model explorations and evaluations based on assertions (1960s), mathematical-logical components (1970s), breakpointing and walk-throughs (1980s), structure-oriented and variability-sensitivity testing (1990s), walkthroughs and extreme conditions testing (2000s), and execution-based and evidence-driven evaluations (2010s). These characteristics pertain to differing artifacts within a simulation, ranging from code to logical components to outcomes. Verifying each of these pieces provides different levels and perspectives of credibility for the correct construction of a simulation. Technological and methodological advances in model building, sharing, and in verification contribute to evolving coverage challenges. Existing challenges pertaining to coverage and increasing model complexities are explored further to assess the current state-of-the-art.

Simulations cannot generally be proven to be completely free of errors; as a result, measures of success are needed to determine if a simulation's implementation can be regarded as credible. Credibility is only provided under the specific conditions tested and does not indicate a lack of errors outside of the tested regions. Pass or fail criteria is generally based on the selected techniques; however, measures can be based on requirements [20, 56], model design [1, 54, 56, 95], the conceptual models [19, 20, 56, 96, 97] as well as from the implementation [1, 19, 20, 54, 56]. The complexity of simulations continues to increase with respect to the number of model components, the number of interaction points, the frequencies of interactions, the creation of hybrid models, the incorporation of big data and social media data, the use of dynamic social network layouts, and dynamic model structure.

An effective experimental design can facilitate the verification phases by reducing the total volume of simulation runs that need to be conducted, clearly capturing the scale and the scope of the simulation being tested, and provide stronger support for any garnered insights. Determining adequate sample sizes in the experimentation phase strengthens model credibility [98] by directly connecting with statistical sample size requirements and revealing how much of the solution space has actually undergone verification. Designs commonly account for sampling, replication, and blocking to represent all portions of the solution space [99]. Constructing a design of experiments helps to reduce the scale of the experimentation through output pair (simulation data points to real work data points) volume consideration to ensure that the minimum required numbers are met and that these numbers are not arbitrarily exceeded without

purpose [100]. Sample sizes also need to be large enough to avoid the introduction of bias into the interpretations [90, 101]. Therefore, it is important to know the coverage limitations and constraints of selected verification techniques when dealing with complex simulations. Identifying techniques in advance that can adequately cover the space within a single simulation run (e.g. the simulation space) as well as the range of possible parameter combinations that can occur across the aggregate simulation runs (e.g. the solution space) can yield significant time savings and provide better support to model credibility.

Current challenges and future direction for verifying simulations of increased complexities and scale:

- **Communicating coverage across individual simulation spaces and the aggregate solution space.** The goal of verification is to identify the existence of errors within a simulation and to help identify the source of the error. Creating a design of experiment to properly search the solution space facilitates greater model credibility while reducing time requirements [102]. Sampling techniques provide great value in reducing the input space that needs to be explored to gain significant insights into the internal operations of a model [103]. Many sampling techniques exist including, Latin hypercube sample (LHS) [104, 105], sampling-based Sensitivity Analysis (SA) [106], and optimization strategies [107, 108]. Extensive literature exists on topics of sampling; however, these discussions are uncommon within the joint context of simulation verification.

In LHS experiments, a subset of input vectors from the solution space are selected and utilized for analysis [103]. For small sample designs, values are selected individually and combined at random. Large sample designs do not need to rely on conditional sampling and can instead pull values directly from a grid design [109, 110]. Many other forms of LHS designs exist that differ in how the sample combinations are generated and how many samples are generated, such as orthogonal and nearly orthogonal designs [99]. Diallo, Lynch [111] and Collins, Seiler [104] both relied on LHS within their experimental designs to reduce the total number of runs needed and save time generating data.

SA analyzes how uncertainty in the model output can be attributed to uncertainty in the model input [112]. This helps to provide insights into the level of dependency that a model output or set of outputs have with respect to one or more input parameters. Therefore, SA evaluations can yield mismatches between the driving forces within a model and the corresponding specifications from the model's design. SA is particularly useful in examining the extreme boundaries of the input space and revealing oddities then produced in the outputs [105]. Additional exploration is required in the presence of correlated inputs to distinguish the distribution of effect between correlated factors [113]. Numerous M&S studies utilize SA to facilitate simulation space and solution space exploration (for example, see Duggan [114], Thiele, Kurth [115], and Hekimoğlu and Barlas [116]).

Verification exist to identify and reveal errors within models of increasing complexities; however, these techniques are rarely presented within the context of sample size requirements and how they can contribute to the overall exploration of the solution space. Instead, this is left to the discretion and experience of the people responsible for designing the experiment. As a result, prerequisite statistical, mathematical, and simulation knowledge requirements are further increased. Statistical debugging using elastic predicates and many-valued labeling functions has been developed for the exploration of simulation using software engineering principles and directly accounts for sample sizes when making verification determinations [82]. Statistical debugging delves into complex simulation interactions without requiring a formal mathematical model specification to identify and isolate locations of potential errors [61, 82, 111]. Diallo, Lynch [111] apply statistical debugging to sets of thousands of simulation

outcomes pulled from an experimental design producing thousands of simulation runs per batch. Utilizing the V&V Calculator [81], adherence to dozens of model specifications were explored in each test and no violations were identified within the final testing set. These studies show potential for addressing scalability, but additional research and techniques are needed. The sampling considerations associated with selected techniques should be clearly understood before attempting to execute an experimental design.

- **Creating and extending techniques to verify temporal-based occurrences and interactions.** Agent-based Models experience challenges in handling networked direct interactions between agents over time [117, 118]. Feldkamp, Bergmann [119] explore the role of knowledge discovery for online analysis of simulation data. They further explore the role that interactive visualization can play in supporting the knowledge discovery process by adding traceability to outcomes across experiment runs [120]. Time series plots are commonplace for displaying quantitative changes over time with a single parameter or feature represented by each line of the plot [121, 122]. Alternatively, sequences of static (e.g. time-sliced) visual artifacts can be provided over numerous time steps to illuminate temporal changes across parameters. For instance, Ahrens, Heitmann [123] utilize series of map-oriented density plots to verify sea surface height errors from comparisons of separate model outputs. An empirical evaluation of the trade-offs between the volume of visual artifacts provided to users and the added value to the verification process is needed.

- **Creating and extending techniques to verify spatial-based occurrences and interactions.** Spatial and abstract (non-spatial) data are handled differently with techniques developed across numerous subfields, such as information visualization, graph visualization, and scientific visualization [124]. Whether the data is univariate or multivariate places additional complexities on the ability to coherently convey spatial information. Focusing on events within the data raises the analysis to a higher level of abstraction and may require additional categorization criteria for exploration [121]. However, the scale of the data being visualized can cause representational errors by distorting relationships [125]. Spatial analytic capabilities focus on spatial (e.g. geographic) relationships [123, 126], spatial density representations [127, 128], patterns [129], and interaction points [130]. For example, consider the value provided by the visual techniques of radar charts and parallel coordinate plots. Radar charts convey time-oriented, multi-dimensional population data to convey trends and trajectories based on the input settings [29]. Parallel coordinate plots using color coded cluster representations have aided knowledge discovery by adding traceability to analyses involving multiple simulation runs' outcomes [120]. These techniques are valuable for revealing targeted insight into multivariate data, but they do not scale well as they result in large volumes of visual artifacts that require manual inspection.

With an increasing variety of social media platforms and easily accessible information posted directly by people about their daily lives, key events, and their likes and dislikes, there are growing possibilities for connecting simulations directly into the "human" component of data. Kavak, Vernon-Bido [31] explore the use of social media data in simulations as sources of input data, for calibration, for recognizing mobility patterns, and for identifying communication patterns. Padilla, Kavak [32] use tweets to identify individual-level tourist visit patterns and sentiment. These information sources can provide new routes towards developing population-based behaviors and rules, but also lead to increased difficulties and needs for effectively verifying simulations utilizing this information. Techniques and methods for identifying errors due to interactions and interconnections between agents based on this type of data are needed.

Lynch, Kavak [128] explore the use of spatial plots and heat maps for identifying suspicious outcomes within ABM execution. Sun, Xu [129] explore pattern formation comparisons to reflect model credibility in the formation of vascular mesenchymal cells and lung development. Courdier, Guerrin [130] use the Geamas Virtual Laboratory (GVL) tool to collect traces of a biomass ABM for analyzing animal wastes management. These traces collect (1) sets of exchanged messages between agents, or (2) a historical accounting of simulation execution per agent or group of agents. Visualization tools inspect these traces and identify interactions that lead to successful agent negotiations. Their visualization filters traces based on specific agents or characteristics. However, the GVL tool becomes unwieldy for analysis of traces once exceeding several dozen agents [130, 131]. Challenges associated with visual representations are further explored in Section 4.3.

- **Creating and extending techniques to verify network-based occurrences and interactions.** Models of increasing structural and behavioral complexities are creating challenges in the ability to clearly interpret and trace the origins of occurrences within simulation runs. This increases the difficulties in determining if system level behaviors within a simulation are the result of the intricacies of these interactions or if they are due to an error in implementation. Such occurrences may be the result of non-linear actions of subgroups or local networks [118, 129], changing model structures over time [28], or the scale of networked or interconnected components [132]. Therefore, techniques need to differentiate between individual-level, subgroup-level, and population-level occurrences [60, 120, 133] to provide traceability between occurrences and model specifications. Network visualizations provide power for data integrations and transformations in revealing ways that can confirm hypotheses or identify new hypotheses by exploring the results of batches of experiments [125]. Network representations provide connecting links between individuals or groups of individuals and can include: social; kindred; online; neighborhood; workplace; religious; and many other types of networks [130, 133–135].

    Agent-based Modeling commonly relies on network structures to handle communications between agents as well as handle agent-agent interactions in order to reveal system level behaviors [136, 137]. For example, verifying human interactions involves checking increasing orders of magnitudes of socially and spatially oriented interactions [30, 34] and increasing quantities of agents and options for modeling decision making [138]. Verifying safety-critical systems deals with identifying issues pertaining to multiple internal layers for self-monitoring, safety checks for maintaining flight paths and altitude [77, 139], and collision avoidance [140]. Epstein [141] develops a mathematical model incorporating, among other factors, social components in order to explore the emergent dynamics of network structure. Dean, Gumerman [142] illuminate the importance of demographic and environmental interactions in recreating histories of sociocultural stability and variation among the Anasazi culture. In both Shults, Lane [134] and Shults, Gore [135], a combination of statistical debugging and visual analytics techniques are utilized to verify the personal-based and environmental-based interactions within their respective models with respect to their intended model designs.

    Node and edge graph visualizations are used to represent structure and weights within network models. Node-link diagrams provide intuitive representations of contact relationships by utilizing basic visual geometries to indicate connections. Sallaberry, Fu [143] show that decomposing a sequence of interactions within a social network yields insight into static network structure as well as dynamic relationship formations and changes over time. Jacomy, Venturini [144] provide an intuitive layout for visualizing relational, network data for facilitating exploration through spatializing networks. Providing visual analytics that are easy to use, easy to understand, and that facilitate exploration is critical for conveying credibility within simulated

networks. For example, Xia, Wang [145] develop a graph-theoretical network visualization toolbox, BrainNet Viewer, to provide ball-and-stick based representations of brain networks. Easily configurable settings allow for adjustments of colors, sizes, and node/edge combinations to allow for comprehensive exploration of models of the connectome information of the human brain. BrainNet Viewer illustrates the need and value offered by easy to use tools that aid in the interpretation of complex networks. Collectively, these works highlight the need for greater depth in easily to apply tools and methods for exploring network dynamics in support of the creation of or confirmation of hypotheses within these complex models.

### 4.3 Developing new feedback styles to support the identification of errors

The prominent concepts from the Verification Corpus indicate a theme of feedback mechanisms pertaining to how insight is provided on the presence of errors. The prominent high-level concepts pertaining to feedback are indicated through *logic* and *behaviors*. Determining how, when, where, and why changes are occurring throughout the execution of simulation provides valuable support in identifying the presence and location of potential errors and faults within a simulation. The defining characteristics across the decades point to a changing reliance on coding and computer-asked (1960s), compile-time and in-program (1970s), behavior-mode and computer-executable (1980s), visually-based and variability-sensitivity (1990s), non-statistical and behavior-sensitivity (2000s), and statechart-assertions and non-bottlenecks (2010s). In general, these characteristics point to programmatic, visual, and statistical feedback to support exploration in searching for errors. Existing challenges pertaining to feedback in support of error identification are explored further to assess the current state-of-the-art.

The mediums used to convey model feedback impact the timeliness and effectiveness of identifying errors based on how easy they are to interpret and how quickly the information can be processed by the user. The defining characteristics of verification reflect an expansion from code-based and numerical-based feedback mediums in the 1960s and 1970s to include visual feedback by the 1980s. Many of the characteristics also focus on the verifying the simulation post execution and point to the use of techniques that have steep learning curves. Verification techniques that require in-depth knowledge of mathematical logic or statistics also serve as a perceived barrier. The use of visual aids are commonly utilized to reduce this burden for the intended audiences, when utilized appropriately [125, 146–149]. Although visual terms were identified within the corpus, few identified studies have explored how to utilize visualizations, integrations of visualization, or combinations of sensory feedback (e.g. tactile, olfactory, or audial senses) to enhance simulation verification. For assisting error identification, Vickers and Alty [150, 151] and Lynch [94] have identified significant value in identifying errors using audial feedback for software and simulations, respectively. Numerous studies have utilized other sensory feedback to enhance the simulation experience [152–154], but they have not explored specific applications of use in supporting the verification process. Visual analytic approaches differentiate techniques' abilities to account for univariate, multivariate, special case, and temporal data types [120, 124], but a deep dive on how and when specific techniques are applicable to existing verification challenge areas within simulations has not yet been conducted.

Current challenges and future directions for providing feedback on verification outcomes:

- **Providing usable, visual feedback of temporal analyses across the solution space.** Time series data provides difficulties in interpretation and visualization as the quantitative aspects are not commonly conveyed along with the visual components and interpretation is left to human perception [121]. Aigner, Miksch [124] suggest time, data, and representation as categorizes analyzing time-oriented data to provide tighter integration of visual and analytical

methods. Multivariate data can lead to challenges in conveying interpretable outcomes that clearly reflect temporal changes due to multiple dimensions, aggregations, or factor interactions [120, 126]. Verification techniques need continued advancements for communicating complex analyses resulting from aggregations of simulation runs and interaction points hidden within layers of interacting components within the simulation. Maintaining the challenges from Section 4.2, sample sizes of temporal analyses need to be conveyed along with the data as techniques may truncate the time series and introduce bias [105]. Verification tools need to facilitate navigation through the combination of the analytical space (the data being analyzed) along with the exploration space (the simulation components producing the data for verification purposes).

- **Avoiding misrepresentation, misalignment, and misuse of feedback.** Visualizations convey information about the simulation; as such, it is important that the selected visualizations do not distort the representation of model outcomes and decrease the usefulness of the model [144, 155]. When incorporating visual components for V&V, several challenges exist for misrepresenting the model. Vernon-Bido, Collins [156] discuss several challenges for visualizing model results, including misrepresenting model components, misrepresenting the magnitude of changes, and poor visualization choices resulting in additional levels of added model complexity. Technical challenges are also important for visualizing simulations, including usability, scalability, and integrated analysis of heterogeneous data [157]. Usability refers to designing visualizations that successfully contribute to the advancement and use of visualization research. Scalability refers to how well visualization tools apply to representing large data sets. Integrated analysis of heterogeneous data deals with visualizing data obtained from various locations in various formats. The use of appropriate layering, separation, and other visual markings can help to avoid unintended spatial interpretations [125].

- **Alternative feedback approaches.** Verification can benefit from research advances targeted at two dimensions of verification: (1) the point in time that verification is being applied; and (2) the sensory feedback mechanism through which results are being communicated. For the first category, technique classifications commonly focus on the level of mathematical formality required of the technique, ranging from informal to formal [1, 2, 7, 10]. Classifying the level of mathematically formality is very helpful for the model builders, simulation testers, or analysis teams that are responsible for selecting and implementing these tests and instrumenting the simulations to facilitate testing. However, these techniques can become inaccessible to people without strong backgrounds in simulation, statistics, or mathematics. As an alternative, it may be more appropriate to develop techniques based on whether a test is intended to be applied to the simulation code, provide indicators during runtime, facilitate input-output analysis, or conduct exhaustive, formal analysis. Runtime approaches may provide better short-term options for gaining acceptance by the M&S community by providing informal, real-time support for revealing potential errors as they occur [77, 158], such as through the use of animations and other visual representations [24, 62, 72, 156] as well as through audial feedback [94]. Emerging runtime approaches help to pinpoint the locations of errors through sound cues [159], spatial plots, and heat maps [128]. Such verification techniques may yield greater value using visual analytics to communicate quantitative information throughout a run while statistical indicators based on population samples may be more appropriate for input-output experimental designs.

Secondly, the type of sensory feedback utilized to convey results can play a significant role in facilitating understanding and acknowledgement in the existence of an error [94]. Visual

feedback is well suited for communicating history [160], spatial information, and referability [161, 162]. However, audial feedback can be more appropriate for communicating flow of control [163, 164], temporal and transient information [165], and conveying information based on information that is not within field of view [166, 167]. A recent survey found a statistically significant difference in the abilities of participants to correctly recognize the presence of an error when using visual and audial feedback together over participants receiving only visual feedback [94]. Further research is needed to explore effective sensory avenues for communicating verification outcomes.

### 4.4 Study limitations

Our study contains limitations. The large number of publications comprising the Verification Corpus gives the appearance that prominence values are always very low; however, this is expected since the size of the Corpus directly affects the calculation of prominence. The number of publications represented each decade differs due to increasing numbers of publications per decade which is in part due to more venues represented over time. We do not examine the differences between the usage of verification between journal and conference venues although this could serve as an interesting examination for future work. The Verification Corpus provides the opportunity to conduct a much deeper, informative comparison between the concepts of verification and validation that could help illuminate their theoretical, practical, and applied commonalities and differences; however, this is outside the scope of the objectives of this work and is the focus of future work. Finally, content analysis describes only what is contained within the body of text, it does not provide explanations for why something does not appear; instead, it is up to the researcher to provide the explanation.

## 5. Conclusion

Academic publications serve as a medium for communicating and disseminating knowledge within disciplines. As such, we construct a Verification Corpus consisting of 4,047 publications and apply content analysis to explore the evolution of simulation verification. Over the past 60 years, we find support that verification foci have shifted from code- and hardware-oriented concepts that are reflective of software engineering and computer science disciplines into M&S-oriented concepts oriented on evaluating the connections between simulation implementation and model design. While foci have changed throughout the decades in response to advances to the discipline, the overall objective of increasing credibility by conveying an error-free simulation remains. Perceptions of high time requirements and a preference towards informal techniques persist over time. Techniques have shifted from primarily mathematically formal and statistically supported analyses towards less formal, less statistically supported analyses. Additionally, we find no evidence of verification techniques, methodologies, or concepts being specific to any single model paradigm. By examining the prominent concepts and defining characteristics of the past six decades, we identify the following categories of verification challenges to inform current researchers, students, and practitioners in M&S, to enable new researchers entering the field, and to suggest areas for continued verification research:

- Advancing methods for conveying simulation credibility while remaining easy to use, interpret, and communicate;

- Providing clear and transparent connections between the application of verification and its overall coverage of the simulation and solution space in response to increasing model complexities; and

- Investigating, creating, and employing new feedback mediums to effectively capture new types of errors and to aid in communicating results.

These challenges reflect the current characterization of verification. They are also representative of analytic concerns resulting from technological advances and increasing data availabilities due to increased connectivity of people through social media sites and applications. While verification and validation provide complementary benefits to the modeling and simulation process, our exploration focuses on highlighting verification's concepts and illuminating the fluidity of changes experienced in its prominent concepts over just ten-year periods. We look forward to the many advances in verification research in response to these challenges as well as the emergence of new challenges as a result of continued technological and societal advances.

## Supporting information

**S1 Data. Configuration of the Leximancer tool.**
(PDF)

**S2 Data. Ranked concept list obtained from the Verification Corpus.**
(PDF)

**S3 Data. Lists of concepts that are prominent alongside verification when we apply content analysis to each decade's publications independently.**
(PDF)

**S4 Data. List of thesaurus terms containing "verification" obtained from conducting separate content analyses on each individual decade within the Verification Corpus.**
(PDF)

## Acknowledgments

We acknowledge and thank all of the societies, organizations, journals, and conferences that permitted access to their publications and allowed for the construction of the Verification Corpus, including: the Society for Modeling & Simulation International (SCS); the System Dynamics Society; the Association for Computing Machinery (ACM); the Journal of Artificial Societies and Social Simulation; Elsevier and Simulation Modelling Practice and Theory; the Simulation Interoperability Standards Organization (SISO); the INFORMS Simulation Society; the Operational Research Society; the Interservice/Industry Training, Simulation and Education Conference (I/ITSEC); the Institute of Electrical and Electronics Engineers (IEEE); and Old Dominion University for providing its students access to numerous publication outlets.

## Author Contributions

**Conceptualization:** Christopher J. Lynch, Saikou Y. Diallo, Jose J. Padilla.

**Data curation:** Christopher J. Lynch.

**Formal analysis:** Christopher J. Lynch, Hamdi Kavak.

**Investigation:** Christopher J. Lynch, Hamdi Kavak.

**Methodology:** Christopher J. Lynch, Saikou Y. Diallo, Hamdi Kavak, Jose J. Padilla.

**Project administration:** Christopher J. Lynch.

**Resources:** Christopher J. Lynch, Hamdi Kavak.

**Software:** Saikou Y. Diallo.

**Supervision:** Saikou Y. Diallo.

**Validation:** Christopher J. Lynch, Hamdi Kavak.

**Visualization:** Christopher J. Lynch.

**Writing – original draft:** Christopher J. Lynch, Hamdi Kavak.

**Writing – review & editing:** Christopher J. Lynch, Jose J. Padilla.

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
