## [Decision Letter · Decision Letter 0]

31 Mar 2020

PONE-D-20-02721

A content analysis-based approach to explore simulation verification and identify its current challenges

PLOS ONE

Dear Prof Lynch,

Thank you for submitting your manuscript to PLOS ONE. After careful consideration, we feel that it has merit but does not fully meet PLOS ONE’s publication criteria as it currently stands. Therefore, we invite you to submit a revised version of the manuscript that addresses the points raised during the review process.

We would appreciate receiving your revised manuscript by May 15 2020 11:59PM. To enhance the reproducibility of your results, we recommend that if applicable you deposit your laboratory protocols in protocols.io, where a protocol can be assigned its own identifier (DOI) such that it can be cited independently in the future. For instructions see: http://journals.plos.org/plosone/s/submission-guidelines#loc-laboratory-protocols

We look forward to receiving your revised manuscript.

Kind regards,

Itzhak Benenson, Ph.D.

Academic Editor

PLOS ONE

Journal Requirements:

Please ensure that your manuscript meets PLOS ONE's style requirements, including those for file naming. The PLOS ONE style templates can be found at http://www.plosone.org/attachments/PLOSOne_formatting_sample_main_body.pdf and http://www.plosone.org/attachments/PLOSOne_formatting_sample_title_authors_affiliations.pdf

Reviewers' comments:

Reviewer's Responses to Questions

**Comments to the Author**

1. Is the manuscript technically sound, and do the data support the conclusions?

Reviewer #1: Yes

Reviewer #2: Yes

2. Has the statistical analysis been performed appropriately and rigorously? 

Reviewer #1: Yes

Reviewer #2: Yes

3. Have the authors made all data underlying the findings in their manuscript fully available?

Reviewer #1: Yes

Reviewer #2: Yes

4. Is the manuscript presented in an intelligible fashion and written in standard English?

Reviewer #1: Yes

Reviewer #2: Yes

5. Review Comments to the Author

Reviewer #1: Authors used the method of content analysis to illustrate the semantic changes to verification based on 4047 publications in last six decades. Challenges and future directions are discussed in three categories. The study of authors can be beneficial for the development of verification process. Here are some comments for authors.

1) For step 2 in Section 2, When searching for a paper containing “verification”, will this paper really focus on verification?. For example, verification term can be in the reference rather the body content.

2) In Table 4, why percentages of verification publications are similar in 1980s and 1990s? Are there any potential reasons for this?

3) In the first paragraph of Section 3.1, can you provide more details about how to come up with these 102 prominent concepts from the Verification Corpus?

4) Right after Table 5, authors mentioned this “the 1970s dispalys …computer netwroks”. I think it is because the rank of “distribution” is higher in 1970s. However, the prominence of “distribution” is actually reduced. For the rank and prominence, which is more important? Can authors give more explanations about this claim?

5) In Table 5, what is HLA here? Please explain it in the content.

Reviewer #2: The paper presents a solid study exploring semantic changes to the concept of simulation verification over the past six decades (1963-2015) with 4,047 peer reviewed Modeling and Simulation (M&S) publications. Content Analysis was done to identify the evolution of the concept. The results were comprehensively discussed based on gathered data in Sections 1-3. The analysis of the data clearly shows the development of the concept through the decades.

However, Section 4. Discussion: Challenges and Future Directions for Simulation Verification does not seem to be based on any content analysis. For example, why did the authors focus on these three areas of ongoing verification challenges-(1) the ever-present need to increase user confidence while facilitating ease of use; (2) the need for increased coverage of verification techniques to handle increasing simulation complexity; and (3) the need to further investigate and contribute advances in feedback mediums for conveying verification results. Was there content analysis done that suggest these are the three areas were most identified in the literature? If yes, please add to complete the study. If not, then suggest to add this disconnect.

6. PLOS authors have the option to publish the peer review history of their article (what does this mean?). If published, this will include your full peer review and any attached files.

Reviewer #1: No

Reviewer #2: No

---

## [Author Response · Author response to Decision Letter 0]

24 Apr 2020

Response to Reviewers for A content analysis-based approach to explore simulation verification and identify its current challenges

We would like to thank both of our anonymous reviewers for their helpful and constructive comments. Our response to each comment is contained below. The rationale behind our response is provided in black text and the corresponding updates implemented within the paper are included in red italics text.

General updates include:

Various grammatical updates to aid clarity.

A renumbering of citations as a result of an added reference.

Reviewer #1: Authors used the method of content analysis to illustrate the semantic changes to verification based on 4047 publications in last six decades. Challenges and future directions are discussed in three categories. The study of authors can be beneficial for the development of verification process. Here are some comments for authors.

1) For step 2 in Section 2, When searching for a paper containing “verification”, will this paper really focus on verification?. For example, verification term can be in the reference rather the body content.

Response: Thank you for this very relevant question. Yes, the focus does remain on “verification”. This is one of the benefits that automated content analysis provides. Applying content analysis to a Corpus minimizes the impact of including articles which mention a targeted term (i) infrequently, (ii) mention it only in the reference list, or (iii) mention it only within an author biography section. This is a result of specifying the term(s) used to define the category of the content analysis (such as “verification”). This category directs the classification and exploration of the text. By setting the term to “verification”, only the terms which appear close to “verification” within the texts are counted within the development of the term matrices and thesauri generated. Therefore, if the term “verification” only occurs within the reference list of an article contained within the corpus, all of its text is ignored during the training and classification processes with the exception of the words contained within two (plus and minus) end of sentence characters within that reference (as end of line punctuation marks serve as distance criteria for what qualifies as related terms). 

This may increase the co-occurrence of terms such as dates, author names, affiliations, publishers, and journal names alongside the term “verification”; however, due to the large size of this corpus, these are not common enough to impact the prominence or frequency of the top concepts identified. In corpus of small sizes, this can be accounted for by manually removing these types of concepts from the identified concepts lists after the initial training phase to specifically remove them from the calculations.

Step 2 of the Methodology section has been updated to provide a short explanation that erroneous papers to “verification” may be included in the corpus as a result of the simple filtering criteria applied, but that this does not impact the prominence or frequency scores.

Addition to Step 2 of the Methodology:

... Note, this simple filtering approach may yield articles that contain the term verification only within their reference lists or author biographies. A benefit of automated content analysis is the assignment of a category for the baseline of the analysis. Since concepts are examined with respect to their relative co-occurrence of the specified category, the remainder of the each text is ignored. Therefore, text within these articles that does not mention verification does not impact the frequency or prominence scores of the identified concepts. The Verification Corpus may provide a slight over-approximation of the number of verification articles, but these do not bias the analysis. Therefore, we do not apply additional manual effort for identifying and removing any such articles from the Verification Corpus.

2) In Table 4, why percentages of verification publications are similar in 1980s and 1990s? Are there any potential reasons for this?

Response: Thank you for your comment. We have decided to eliminate Table 4 as it created ambiguities and its content does not critically contribute to the overall theme of the paper. Additionally, this table appears to add more questions than value in its current form. Table 4 has been removed and the paragraph that precedes it has been modified. Table numbers have been updated as a result. Paragraph 2 of Section 3 now reads:

Roughly 19.36% (4,047 of the 20,905 articles) of the overall articles pulled from the selected M&S venues form the Verification Corpus. However, when looking at the venues which contributed papers that were did not just contain verification this percentage drops to 17.02% (3,457 out of 20,315 articles). This is not necessarily a negative observation as these articles range from M&S applications and case studies to theory, methodology, and technology. A one in five reporting rate may be due to a lack of standardized approaches for conducting and reporting verification activities across modeling paradigms and methods. This finding supports the argument that verification is underutilized within the community and that simpler means for conducting and communicating its results are needed (7, 8, 44, 54).

3) In the first paragraph of Section 3.1, can you provide more details about how to come up with these 102 prominent concepts from the Verification Corpus?

Response: The 102 identified concepts of the Verification Corpus include all concepts within any decade with a prominence value greater than 1.0. This does result in concepts showing prominence scores less than 1.0 within various decades, but always greater than 1.0 in at least one decade. This discussion has been added to the beginning of the second paragraph of Section 3.1.

The formation of the concept list includes each concept that has a prominence score of at least 1.0 within any decade. This results in concepts showing prominence scores less than 1.0 within various decades, but always greater than 1.0 in at least one decade. For exploration, we examine...

4) Right after Table 5, authors mentioned this “the 1970s displays …computer networks”. I think it is because the rank of “distribution” is higher in 1970s. However, the prominence of “distribution” is actually reduced. For the rank and prominence, which is more important? Can authors give more explanations about this claim?

Response: These are rankings with respect to “verification” as the defined category of the content analysis, as described in Section 2. Prominence values are relative to “verification” within each decade based on the total occurrences of each term each decade. As a result, the magnitude of a concept’s prominence is expected to differ each decade and we discuss the ranking of the concepts as a comparison point across decades. The paragraph preceding Table 5 has been updated to reflect this.

The verification concepts are explored with respect to their prominence rankings in each decade. Prominence is utilized for this exploration because prominence scores are relative to “verification” based on the total occurrences of each related term each decade. As a result, the range in the magnitude of concept prominences is expected to differ each decade.

5) In Table 5, what is HLA here? Please explain it in the content.

Response: HLA stands for the High Level Architecture. HLA is an IEEE standard for distributed simulations that is a recommended standard by NATO for this purpose. The Department of Defense’s (DoD) Defense Modeling and Simulation Office (DMSO) developed the HLA in the mid-1990s to serve as a DoD standard for meeting simulation interoperability requirements. As this standard deals with the proper implementation and execution of simulation federates and was developed by the DoD (one of the largest sources of developed simulations), HLA is an appropriate concept to observe alongside verification during this time. 

A footnote has been added to the table (now numbered Table 4) to reflect what HLA stands for and a reference has been added for support.

aThe High Level Architecture (HLA) is an IEEE Modeling and Simulation Interoperability Standard developed by the Defense Modeling and Simulation Office (DMSO) and adopted by NATO (70). The HLA facilitates specifying and exchanging information when creating a simulation by federating simulations.

Reviewer #2: The paper presents a solid study exploring semantic changes to the concept of simulation verification over the past six decades (1963-2015) with 4,047 peer reviewed Modeling and Simulation (M&S) publications. Content Analysis was done to identify the evolution of the concept. The results were comprehensively discussed based on gathered data in Sections 1-3. The analysis of the data clearly shows the development of the concept through the decades.

However, Section 4. Discussion: Challenges and Future Directions for Simulation Verification does not seem to be based on any content analysis. For example, why did the authors focus on these three areas of ongoing verification challenges-(1) the ever-present need to increase user confidence while facilitating ease of use; (2) the need for increased coverage of verification techniques to handle increasing simulation complexity; and (3) the need to further investigate and contribute advances in feedback mediums for conveying verification results. Was there content analysis done that suggest these are the three areas were most identified in the literature? If yes, please add to complete the study. If not, then suggest to add this disconnect. 

Response: Yes, these challenges are based on the content analysis. We agree that the connections between the analysis and the identification of challenges were not clearly communicated. The challenge areas are identified based on the verification concepts identified in Section 3.3 along with the verification definitions presented in Section 3.4 based on the content analyses of each decade. The three identified challenge areas are based on common themes observed throughout the progression of the decades. Additional literature is reviewed in Section 4 to illuminate the state-of-the-art with respect to each of these challenge areas. This also supports the current relevance of these challenges within the field. 

A number of additions have been added to properly provide this transition. Step 7 of the Methodology, the final row of Table 3, the final paragraph of Section 4, the introductions to Sections 4.1, 4.2, and 4.3, and Section 5 have been updated to make this point more transparent and aid in clarity.

Addition to Table 3:

Identify challenges and future directions for simulation verification research (Section 4) Explore common themes within the concepts and definitions pertaining to verification over time to identify existing challenges. Then, conduct a literature review to reflect the state-of-the-art. The evolution of concepts and defining characteristics identified from Sections 3.3 and 3.4

Addition to Step 7 of Methodology section:

...defining characteristics from Step 6 and the concepts identified from Step 5. To reflect the current state-of-the-art of these challenge areas...

Addition to the final paragraph of Section 4:

From the evolution of concepts identified in Section 3.3 and the defining characteristics identified in Section 3.4, we identify...

...We explore the current state-of-the-art with respect to each of these challenge areas by conducting a literature review of recent publications to identify their current challenges. The goal is to inform current researchers, students, and practitioners on existing challenges, to enable new researchers entering the domain of M&S, and to illuminate avenues for future verification research.

Addition to Section 4.1:

The prominent concepts from the Verification Corpus indicate a sustained theme of conveying confidence and facilitating ease of use. As this is a driving premise of verification, this is expected within the analysis. However, the changing characterizations of verification over time reflect thematic differences within this theme. Concepts of test, requirements, analysis, design, development, and results reflect the traditional verification aspects of determining that a simulation has been implemented correctly. The defining characteristics across the decades reveal insight into shifting focal points within this theme, such as investigating (1960s), concurrency and fault-free (1970s), substantiating and accountability (1980s), substantiating and assuring (1990s), truthfulness and accreditation (2000s), and accreditation and explainable (2010s). The pursuit of confidence has ranged from searching for concurrency errors and identifying software faults to substantiating adherence to requirements and design to establish credibility and truth. Existing challenges pertaining to confidence and ease of use are explored further to assess the current state-of-the-art.

Addition to Section 4.2:

The prominent concepts from the Verification Corpus indicate a theme of verification coverage with respect to model complexities. Concepts of logic, software, techniques, process, and behavior reflect differing levels and requirements of simulation space coverage in examining simulation models. Defining characteristics within the decades include model explorations and evaluations based on assertions (1960s), mathematical-logical components (1970s), break-pointing and walk-throughs (1980s), structure-oriented and variability-sensitivity testing (1990s), walkthroughs and extreme conditions testing (2000s), and execution-based and evidence-driven evaluations (2010s). These characteristics pertain to differing artifacts within a simulation, ranging from code to logical components to outcomes. Verifying each of these pieces provides different levels and perspectives of credibility for the correct construction of a simulation. Technological and methodological advances in model building, sharing, and in verification contribute to evolving coverage challenges. Existing challenges pertaining to coverage and increasing model complexities are explored further to assess the current state-of-the-art.

Addition to Section 4.3:

The prominent concepts from the Verification Corpus indicate a theme of feedback mechanisms pertaining to how insight is provided on the presence of errors. The prominent high-level concepts pertaining to feedback are indicated through logic and behaviors. Determining how, when, where, and why changes are occurring throughout the execution of simulation provides valuable support in identifying the presence and location of potential errors and faults within a simulation. The defining characteristics across the decades point to a changing reliance on coding and computer-asked (1960s), compile-time and in-program (1970s), behavior-mode and computer-executable (1980s), visually-based and variability-sensitivity (1990s), non-statistical and behavior-sensitivity (2000s), and statechart-assertions and non-bottlenecks (2010s). In general, these characteristics point to programmatic, visual, and statistical feedback to support exploration in searching for errors. Existing challenges pertaining to feedback in support of error identification are explored further to assess the current state-of-the-art.

The end of Paragraph 1 of Section 5:

...By examining the prominent concepts and defining characteristics of the past six decades, we identify the following categories of verification challenges to inform current researchers, students, and practitioners in M&S, to enable new researchers entering the field, and to suggest areas for continued verification research:

• Advancing methods for conveying simulation credibility while remaining easy to use, interpret, and communicate;

• Providing clear and transparent connections between the application of verification and its overall coverage of the simulation and solution space in response to increasing model complexities; and

• Investigating, creating, and employing new feedback mediums to effectively capture new types of errors and to aid in communicating results.

All co-authors have approved these revisions.

-Christopher J. Lynch, Corresponding Author

---

## [Editor Report · Decision Letter 1]

27 Apr 2020

A content analysis-based approach to explore simulation verification and identify its current challenges

PONE-D-20-02721R1

Dear Dr. Lynch,

We are pleased to inform you that your manuscript has been judged scientifically suitable for publication and will be formally accepted for publication once it complies with all outstanding technical requirements.

With kind regards,

Itzhak Benenson, Ph.D.

Academic Editor

PLOS ONE
---

## [Editor Report · Acceptance letter]

30 Apr 2020

PONE-D-20-02721R1 

A content analysis-based approach to explore simulation verification and identify its current challenges 

Dear Dr. Lynch:

I am pleased to inform you that your manuscript has been deemed suitable for publication in PLOS ONE. Congratulations! Your manuscript is now with our production department. 

With kind regards,

on behalf of

Professor Itzhak Benenson 

Academic Editor

PLOS ONE